# Political prioritization and the competing definitions of adolescent pregnancy in Kenya: An application of the Public Arenas Model

**Maricianah A. Onono**[1]*, **George W. Rutherford**[2], **Elizabeth A. Bukusi**[1], **Justin S. White**[2,3], **Eric Goosby**[2], **Claire D. Brindis**[2,3,4]

**1** Kenya Medical Research Institute, Centre for Microbiology Research, Nairobi, Kenya, **2** Institute of Global Health Sciences, University of California, San Francisco, San Francisco, California, United States of America, **3** Philip R. Lee Institute for Health Policy Studies, University of California, San Francisco, San Francisco, California, United States of America, **4** Adolescent and Young Adult Health National Resource Centre, San Francisco, California, United States of America

* maricianah@gmail.com

## Abstract

### Background

Approximately one in every five adolescent girls in Kenya has either had a live birth or is pregnant with her first child. There is an urgent need to understand the language and symbols used to represent adolescent pregnancy, if the current trend in adolescent pregnancy is to be reversed. Agreement on the definition of a societal problem is an important precursor to a social issue's political prioritization and priority setting.

### Methods

We apply the Public Arenas Model to appraise the environments, definitions, competition dynamics, principles of selection and current actors involved in problem-solving and prioritizing adolescent pregnancy as a policy issue. Using a focused ethnographic approach, we undertook semi-structured interviews with 14 members of adolescent sexual reproductive health networks at the national level and conducted thematic analysis of the interviews.

### Findings

We found that existing definitions center around adolescent pregnancy as a "disease" that needs prevention and treatment, socially deviant behaviour that requires individual agency, and a national social concern that drains public resources and therefore needs to be regulated. These competing definitions contribute to the rarity of the topic achieving traction within the political agenda and contribute to conflicting solutions, such as lowering the legal age of consenting to sex, while limiting access to contraceptive information and services to minors.

**Data Availability Statement:** Data cannot be shared publicly because this study was conducted with approval from the Kenya Medical Research

Institute (KEMRI) Scientific and Ethics Review Unit (SERU), which requires that we release data from Kenyan studies (including de-identified data) only after they have provided their written approval for additional analyses. As such, data for this study will be available upon request, with written approval for the proposed analysis from the KEMRI SERU. Their application forms and guidelines can be accessed at https://www.kemri.org/seru-overview. To request these data, please contact either the authors or the KEMRI SERU at seru@kemri.org

**Funding:** No specific funding provided.

**Competing interests:** The authors have declared that no competing interests exist.

## Conclusion

This paper provides a timely theoretical approach to draw attention to the different competing definitions and subsequent divergent interpretations of the problem of adolescent pregnancy in Kenya. Adolescent reproductive health stakeholders need to be familiar with the existing definitions and deliberately reframe this important social problem for better political prioritization and resource allocation. We recommend a definition of adolescent pregnancy that cuts across different arenas, and leverages already existing dominant and concurrent social and economic issues that are already on the agenda table.

## Introduction

The health and well being of adolescents in Sub-Saharan Africa is of critical importance to the future of Africa and the achievement of the 2030 Sustainable Development Goals. In Sub-Saharan Africa, adolescents (10–19 years) make up 23 per cent of the region's population [1]. When given the right policies and investments, this ever-expanding youth bulge represents an opportunity to reap a demographic dividend, promote gender equity, accelerate economic growth and reduce poverty [2]. Unfortunately, this benefit may not be reaped if one in five adolescent girls are pregnant and unable to complete their education and acquire the necessary skills required for gainful economic activity [3–6]. In countries such as Kenya, approximately 18% of girls between 15–19 years have begun childbearing or already have a child, and 13,000 teenage girls drop out of school every year due to pregnancy [7]. In fact, if all adolescent girls completed secondary school and if adolescent mothers were employed, instead of becoming pregnant, the cumulative effect could add an estimated 3.4 billion US. Dollars to Kenya's gross income every year [8]. The demographic dividend, therefore, can only occur if Sub-Saharan Africa countries improve the legal and policy environments to empower and enable adolescents, and in particular girls, to attain the highest standard of health including, sexual and reproductive health (SRH) [9].

Problem definition is the process of how networks of individuals or organizations generate consensus on what the problem is and how it should be addressed. Problem definition is an important precursor to political prioritization and agenda setting, that is, the process that determines which issues political actors pay serious attention to at any given time [10]. Within this process, unless a "difficulty" is converted into a stated problem, it remains embedded in the realm of nature, accident and fate [10]. Moreover, the complexity of the problem and the potential consequences of divergent interpretations are highly pertinent to adolescent pregnancy, given that adolescent sexuality cuts across national, community, household and individual boundaries. Ultimately, the problem definition that comes to dominate public discourse has profound implications for future solutions in terms of policy formulation and resource allocation. It is, therefore, imperative that we explore and understand the language and symbols that are used to represent adolescent pregnancy if we are to reverse its current trends.

The development and maintenance of an issue on the public agenda is a topic that has been the centre of discussions for many political scientists. Often the topics that make it to the public agenda are neither the largest in magnitude nor the most grave [11]. For example, while women comprise more than half of the population of Africa and bear the brunt of adverse SRH outcomes, prior to 2003, the African Union Charter on Human Rights had only one of its 68 articles that specifically referred to women and girls [12]. Furthermore, this single article

combined the rights of women and girls with the rights of other vulnerable groups, such as the disabled, children and the elderly. In 2011, the World Health Organization released guidelines on preventing early pregnancy and poor reproductive outcomes among adolescents in developing countries, and, while many scholars, funding agencies and national governments adopted the guidelines and were optimistic about the future of the adolescent girl [13, 14], adolescent pregnancy has remained a pernicious problem.

## Theoretical underpinning: Public Arena Model

For this study, we utilize the Public Arenas Model, developed by Hilgartner and Bosk [11]. This theory explains why and how certain social problems are able to rise to prominence compared to others and why some later lose their place on the public agenda, while others persist [11]. The main assumption of this theory is that public attention is a scarce resource, which is allocated through competition in public arenas. In this paper, we apply the six elements of the Public Arenas Model: 1) the public arenas where the issue of adolescent pregnancy is discussed; 2) the carrying capacity of the arenas; 3) the dynamics of competition; 4) the principles of selection; 5) communities of operatives; and 6) feedback mechanisms (Table 1). Although the Public Arena Model was developed in 1988, its inclusion of ecological concepts such as competition, selection and adaption within cultural and political settings makes it appropriate for examining value laden social issues such as adolescent SRH in Kenya.

## Materials and methods

We employed a focused ethnographic methodology to collect data for this study [15–17]. This methodology was chosen in this study for its ability to allow for exploration as well as understanding how adolescence and adolescent SRH are perceived and constructed in a real-world context [18]. In this analysis we adhere to the consolidated guidelines (COREQ) for reporting qualitative research (see S1 File) [19].

## Study setting

The study was done in Kenya. Kenya has shown progressive leadership in the adolescent SRH matters through adoption of favorable international and regional policies and legal

**Table 1. Elements of the Public Arena Model.**

| Element | Definition |
|---|---|
| Institutional/public arena | The environment where social problems compete for attention and grow or diminish. |
| Carrying capacity | The number of social problems that can be entertained within any particular arena. Each arena has finite resources and has both individual "selfish" and altruistic goals. |
| Dynamics of competition | Issues compete against each other and also within their own definitions. |
| Principles of selection | Institutional, political and cultural factors that influence the probability of survival of competing problem formulations. These include: a) competition for prime space, b) dramatization of the issue, c) cultural pre-occupations and mythic themes in the society, d) prevailing political biases, e) carrying capacity of the different arenas and f) institutional rhythms, such as election cycles. |
| Communities of operatives | The networks of persons or organisations that promote and attempt to control particular problems. |
| Feedback mechanisms | The patterns of interactions among the communities of operatives as they crisscross through the different arenas. |

frameworks that promote adolescent SRH [12, 20–25]. However, the nation continues to struggle with high adolescent pregnancy rates.

## Recruitment

We used purposive and snowball sampling to identify and recruit participants. Purposive sampling was used to consciously identify potential participants. A list of potential participants was developed and prioritized according to the following criteria: job position that was previously or currently held, expected expertise and knowledge that they possess regarding SRH. Subsequently, we used snowball sampling by asking interviewees to suggest others who have contributed or influenced the processes and conduct the interviews with them. Lastly, we used theoretical sampling to select further types of interviews based on what would advance insight into the issue of problem definition. Eligibility criteria included state and non-state policy actors in Kenya who are involved in the adolescent SRH policymaking process. State actors that were targeted included senior government officials from the ministries of health, youth and gender affairs, devolution and planning, and education. We excluded officials from the sub-national governments since within Kenya's devolved health system; policymaking is a national function. Participants were contacted via telephone, given a brief overview of the study, and asked if they were willing to participate.

Our ethnographic methodology precluded *a priori* sample size estimation; however, for planning, we estimated that we would need to conduct in-depth interviews with approximately 15–20 individuals before reaching a data saturation point. Emphasis was placed on ensuring that there were equal numbers across a range of state and non-state actors. Recruitment continued until saturation was reached.

## Data collection

We collected primary qualitative data using semi-structured in-depth interviews (IDIs). Interviews were conducted in English, lasted approximately 90 minutes, and were digitally recorded and transcribed verbatim by a masters level qualitative researcher. In line with an ethnographic approach, we also documented reflective field and memoranda to keep track of any emerging theoretical insights throughout the data collection process. The in-depth interview guide used in the study is included in S2 File and included questions on 1) the current state of priority for adolescent SRH in the health agenda of Kenya, 2) who holds significant influence on how adolescents and adolescent SRH are defined and addressed, and 3) how adolescent SRH can be re-framed to political leaders in order to generate political support and bring it to the agenda table.

**Data management and analysis.** All interviews were conducted in a private location at the participant's discretion. A professional transcriber prior to analysis transcribed interview transcripts. The interview transcripts were read and reread by the lead investigator and a social scientist trained in qualitative methods to identify emerging codes and categories The data were analyzed using a theory-informed thematic analytical approach [26] using Dedoose qualitative software. Transcripts were coded paragraph by paragraph by the two researchers. Consistency of coding between the two researchers was established by initially coding the same transcripts and through frequent discussion between coders until consistency was fully established. An effort was made to ensure that the emergent codes and themes remained close to both the data and relevant literature [27, 28]. Throughout data collection and analysis, we practiced reflexivity by continually examining our own biases as former and existing members of the national adolescent technical working group, preferences, and theoretical perspectives

and how those factors played a role in our understanding and interpretation of the processes and data we were analyzing [18].

**Ethical considerations and protection of human subjects.** The research was reviewed and approved by the Scientific and Ethics Review Unit (SERU Study 3738) at the Kenya Medical Research Institute (KEMRI) and the Committee for Human Research of the University of California, San Francisco (UCSF). All participants (Table 2) provided written informed consent prior to the interview being conducted. The digital audio recording of the in-depth interviews were not initiated until after the informed consent process was complete, the participant had agreed to the recording, and any initial introductions that might include identifying information had been completed. Participants were not reimbursed for participating in the study.

# Results

## The public arenas and competing definitions

The environment where social problems compete for attention and grow or diminish is known as the public arena [11]. Within Kenya, the main arenas where adolescent pregnancy is discussed and defined are the health sector, the executive and legislative branches of government, religious and cultural groups and non-governmental and civil society organizations. Because there are many different players, within each of these arenas adolescent pregnancy is constructed differently. For example, *the health arena* defines adolescent pregnancy from a biomedical perspective as a disease that requires prevention, treatment and monitoring. As shown in the quote below, within this arena, adolescent pregnancy is defined as being undesirable, unplanned or unwanted and is associated with major social problems, including persistent poverty, school failure, child abuse and neglect, health issues and mental health issues [29, 30] including higher risk of maternal mortality [31]. This definition, therefore, emboldens the health arena's claim for programmes and research to prevent and manage adolescent pregnancy. Such programmes focus on increasing an individual's unfettered access to contraceptive information and expanded mix of contraceptive choices [32–40].

**Table 2. Institutional affiliations of subjects.**

| IDI No. | Name | Type of Actor |
|---|---|---|
| 1. | Ministry of Health: | Government |
| 2. | Ministry of Health: | Government |
| 3. | United Nations Population Fund | International Development NGO |
| 4. | Population Council | International NGO |
| 5. | Sexual Reproductive Health and Rights Alliance | Civil Society Organization |
| 6. | Kenya Medical Training College, Nairobi | Government |
| 7. | PATH international | International NGO |
| 8. | Inter Religious Council of Kenya | Civil Society Organization |
| 9. | Ministry of Youth | Government |
| 10. | National Council for Population and Development | Government- State Corporation |
| 11. | National AIDS and STI Control Program | Government (Ministry of Health) |
| 12. | JHPIEGO | International NGO |
| 13. | Youth Counselor | Youth representative |
| 14. | National Organization of Peer Educators | Civil Society Organization |

"*Between the age of 15 to 20 years that is when most pregnancies occur. If you are going to reduce teenage pregnancy, it means that you are going to reduce issues of maternal deaths. This is because, one, physiologically they are not mature. Point number two they are dependent on their parents. They are not independent. So issues like on how they are going to support themselves, how they going to support their children will also come in because we are also looking at neonatal, prenatal, child health thereafter. So it is very important that we actually put enough resources to make sure that we delay pregnancies. Once the pregnancies are delayed a bit for some years then it makes a difference.*" (*Reproductive maternal health services; IDI_2*)

In contrast, *the political arena*, that is, the executive and legislative branches of government, defines adolescent pregnancy as a national social concern that needs to be addressed for the country to reap a demographic dividend as explained by an official from the ministry of youth below. The prevailing narrative is that adolescent mothers are less likely to complete the education necessary to qualify for a well-paying job [13, 14], and, if all adolescent girls completed secondary school and were employed, instead of having a child, the cumulative effect could add 3.4 billion US dollars to Kenya's gross income every year [8]. Unfortunately, this definition avoids the upstream social structural factors, such as poverty, increased urbanization or non-enforcement of school re-entry programmes after pregnancy, and results in misplaced corrective actions that include restrictive laws that criminalize sex or conservative laws that that do not allow contraception for adolescents, for fears that contraception use leads to promiscuity [9, 41]. Kenya, for example, has recently proposed to cut the age of consensual sex from 18 to 16, but does not permit this age group to access information and use contraception or sexually transmitted infection prevention services [42].

"*We are talking about the issue of demographic dividend which cuts across all the youth; I think you have heard about the youth demographic dividend. . .. I can say this is where we are; these are the number of youth at school, these are the numbers that fall out due to pregnancies, these are the ones that stay out at school due to menstruation and therefore this is how it affects them; menstruation issues for girls, abortions, STIs, those that are being married off, the ones that are maybe not coming back to school after they have been circumcised or they are married when they are very young and how much it affects the community. If these problems were mitigated, then maybe in 20 or 10 years down the line, we monitor the school completion rate and maybe after another 5 years after they are through with the university and they are employed, we monitor how much they contribute to the society and development*"(Ministry of Youth; IDI_9)

The *cultural arena* defines adolescent pregnancy as an antisocial tendency or a problem of weak morals or agency. However, this definition does not hold when motherhood happens as highlighted in the quote below. Some cultures such as the Turkana or the Maasai in Kenya will accept pregnancy as long as the girl is married [43], and other Kenyan cultures see adolescent pregnancy as a form of rational adaptation in which girls choose to become pregnant because they believe that a pregnancy will lead to marriage [44]. Within this context, some adolescent pregnancies are not problematic even if outlawed by the constitution. In fact, 23% of Kenyan girls are married before their 18th birthday, and 4% are married before the age of 15. Invariably, these cultures have higher adolescent pregnancy rates (~40%) when compared to the national average (18%) [7, 43].

"*Culture is a barrier because issues of early marriages are encouraged in some communities and that leads to early teenage pregnancies which will affect the health of the mother; issues of female genital mutilation. There is the cultural issues that once you are circumcised, now you are ready, you are a woman, you can be involve in all manner of things. There is also the issue of religion; some churches do not want anything to do with family planning or the sexual reproductive health*" (NCPD; IDI_10).

**Carrying capacity of public arenas.** Each public arena has a different carrying capacity, that is, limited resources that restrict the number of social problems that it can handle over a period of time (Table 3). This means that the different social problems must compete both to enter into and remain visible in an arena. The amount of carrying capacity that a particular arena has is socially constructed [45]. At any given point, arenas are juggling several issues within the public sphere. Each arena also struggles with resource management and how to maintain relevance. The theoretical notion of a carrying capacity does not allow for increased space or surplus compassion. It is a zero-sum game. When the visibility of one issue increases, that of another issue decreases. One respondent lamented on how easily "prioritized" problems quickly lost visibility in the public arena.

"*And then again, something happens, for instance, it's just the other day were talking about pregnancies; alarm that so many girls are giving birth during the exams and all that. . . It's sad that Kenyans forget things so easily. When things happen, we make noise and then it cools down and we forget about it*"(Ministry of Youth; IDI_9).

Globally, there is no health system, poor or wealthy, privately or publicly financed, that can afford to provide all possible health services for the people it serves [46]. The health arena, therefore, has to balance a finite budget and a limited number of skilled health care workers and technical capacity.

"*You know when I look on like the Abuja declaration which says that every government should set aside like 15% of their GDP for health, has Kenya done that? No. . .. I don't think Kenya we are there. We are not there yet in terms of the priority we give to health. I think the*

**Table 3. Carrying capacities and resource constraints of different arena and operatives.**

| Unit of analysis | Resource constraints |
|---|---|
| **Public arena** | |
| Donor agencies | Total budget, other programs being supported, time, local or global cost of action |
| Parliamentary health committees | Time, number of staff, budget, political cost of action |
| Civil society organizations/ non-profits/ other non governmental organizations | Time, number of staff (paid and volunteer), budget, political cost of action |
| **Operatives** | |
| Politicians | Time (personal and within electoral cycle), number of staff, budget, political cost of action, media slots (paid and free) |
| Reporters | Time, budget, energy, political and social capital with editors |
| Academicians/researchers | Time, free media slots, social and political capital, funding, capacity to communicate |
| Legal organizations | Time, free media slots, social and political capital, funding, capacity to communicate |
| **Members of the public** | Money, time, surplus compassion, social costs, other problems |

*Ministry of Finance and the Ministry of health are not aligning the finances to the need... There is no sustainability plan to ensure that these programs run. So, lack of sustainability plans to ensure that some of the programs that are targeting ASRH are sustained and they are continuous so they die off along the way; or they start on a very high note then die"(Religious organization; IDI_8)*

"*There are not enough resources to employ enough staff such that you have some set aside for ASRH*"(*Ministry of Youth; IDI_9*)

On the other hand non-governmental and civil society organizations have constraints in the number of staff that they can allocate to focus on adolescent pregnancy, the levels of funding available that can be leveraged, the amount of time that can be allocated, as well as the pernicious issue of the political cost of their sustained action on this matter and how this bodes with their existence both at local and global level. For example while there are no federal funds supporting abortion in Africa, the recent "global gag rule" by the United States placed a restriction on funding and how it can be spent and can lead to non-governmental organizations shutting down and further reducing access to contraceptives [47, 48]. Furthermore, the reduction in funding to the United Nations Population Fund by the US government means that contraceptive commodity security will be further jeopardized.

"*You have donors suffering from donor fatigue and telling you that their governments want to focus on remodeling so the US are telling you Americans first. The European are decreasing the level of funding they are giving us because they say after 50years of consistent support to us we should have owned our issues and we should be financing them which is true if you look at it in all fairness, and they would like to move on to other issues, may be climate change and things like that. The donor world changes*" (*International NGO; IDI_7*)

Within the *political arena*, there is often a limited amount of time available during parliamentary sessions to debate issues, and the political costs and social capital of certain decisions may be perceived as carrying additional "political" costs. Moreover, the regular election cycles keep elected government officials in a perpetual state of preoccupation with staying in office and maintaining power. As a result of this preoccupation, the political class has minimal surplus compassion for issues that do not have "celebrity status" in the community or for populations that are unlikely to swing the vote in their favour at the next election.

"*Another problem that we have as a country is whereby I'm Governor X; I would say this is the direction we are taking as a county, this is our County Integrated Development Plan and we've agreed this is the direction we are taking; Governor Y comes in and feels like those projections you've made are Governor X's and now we are going to do mine; so there is no continuity, there is no buy in of what had been initially planned as much as the community had adopted it; maybe there was even community participation but now you have to start fresh community participation forums. There is a lot of wastage of resources un-necessarily*"(*Ministry of Youth; IDI_9*)

**Principles of selection and dynamics of competition.**    The principles of selecting an issue are the factors that influence the probability of its survival (Table 1). There are two levels of competition. First, there is competition to define an issue as being "worthy" of societal attention. Secondly, issues (and their advocates) themselves compete against others in order to achieve prominence, attain valuable resources and be (and remain) on the agenda. For example, a national mobile phone-based survey, identified the top problems people wanted to have

discussed in Kenya in 2019 as: corruption, the high cost of living, unemployment, poor leadership, poverty, hunger, tribalism, poor infrastructure, terrorism and crime [49]. Despite the apparent magnitude, urgency and impact on the aforementioned list of problems, poor health and lack of access to quality health services (including SRH) have not been acknowledged nor deemed high enough to be on the "discussion table". In fact, in 2017, the political campaign for the Kenyan presidency and the resultant political instability dominated the public arena; consequently, a concurrent doctors' and nurses' strike that paralysed the public health sector was completely marginalized over a 120-day period and barely mentioned in the mainstream media.

The size of the carrying capacity within the arenas determines the amount of competition faced by advocates and different issues. Arenas with small carrying capacities, such as the political arena, have more intense competition. Each problem, therefore, needs to be able to be dramatic and demonstrate novelty in order to capture an audience. Novelty involves the use of symbols to dramatise problems. This is particularly important for problems that can be normalised such as adolescent pregnancy within the context of child marriage.

> "*In some areas, because those are married girls, in as much as they are adolescents, that is somebody's wife so there is little you can do*" (Ministry of Health; IDI_1)

Cultural preoccupations and political biases can also de-dramatise the burden of adolescent pregnancies and normalise it. As described earlier, while adolescent pregnancy outside marriage is frowned upon, adolescent pregnancy in and of itself is accepted [50]. These cultural preoccupations, which are often enforced by religious beliefs, keep adolescent pregnancy at the margins of the public arena, falling into an area of implied acceptance, rather than attempting more controversial solutions, such as access to contraceptive services, which become symbolic of sexual activity that is perceived to be "condoned".

> "*One of the things we are learning in adolescent sexual reproductive health as programmers is that when you frame it in the context of population, politicians are not interested. They want numbers, they want people to have many children; which is completely contrary*"(International NGO; IDI_7)

Lastly, political interests can affect the very emergence of adolescent pregnancy as a social problem. In many cases, adolescents (10–19 years) are considered to be dependents [51, 52]; and until they reach the age of 18, have little power and influence insofar as their ability to vote and participate in the political process and to contribute to the economy. In this regard, many adolescent issues in many SSA countries are in a state of "politically enforced neglect" with politicians and policymakers focusing on groups and problems that earn them the most political mileage, such as investing in infrastructure.

> "*In the area politics, politicians will not say certain things because when they say so, they can lose votes because they will be having certain stands against the community. There are community leaders who are very strong and influential even in choosing who will be their political leader. So some of these political leaders will accept certain things in certain forums but when they go home it is a different ball game*" (UN agency; IDI_3)

**Communities of operatives and feedback mechanisms.** The community of operatives refers to the groups and individuals from different sectors of society that publicly present potential problems and whose channels of communications crisscross different arenas. These operatives

come from different arenas and invariably have different goals and varying degrees of perceived power. Table 3 lists the different communities and the resources they need. Operatives are usually very familiar with the principles of selection and are able to frame their issue in politically correct rhetoric. Interactions and feedback within the community of operatives help frame and reframe social issues and can determine how long an issue remains in the public arena.

> "*If you ask me, the way we frame our messages has to be culturally appropriate, age sensitive and we need to also work with the politicians, who are going to be our champions on how to share those messages. Mmmh, at the end of the day, health relies mainly on governance and leadership and political will. So if a certain issue has been picked up by the political arm then it becomes easy to roll it out. This is because it means facilitation such as technical support, finances, everything will be provided. So mainly it is influenced by whether the issues have been taken up politically*"(International NGO; IDI_7).

## Discussion

Priority setting for health interventions is one of the most challenging and complex issues faced by health policy decision-makers all over world [53, 54]. How adolescent pregnancy is defined has a powerful influence on public officials and helps shape policy design, selection of acceptable interventions and resource allocation. In this paper, we demonstrate that the competing and divergent definitions of adolescent SRH among various operatives contribute to its lack of prioritization by the political class. In addition single definitions may not get much traction. For example, defining adolescent pregnancy only as an individual moral issue relegates the solution to one of individual agency and which the state perceives that they can have little influence. Alternative and cohesive public positioning of adolescent SRH are therefore required. Lastly, it is imperative that particularities of each public arena and the actors involved in each of the arenas be analysed and leveraged. Below we provide some recommendations.

First, we propose more broad and heterogeneous composition of adolescent SRH stakeholders. It is unlikely that a single arena or a non-collaborative community of operatives can increase the public concern and policy importance of adolescent pregnancy. Communities of operatives that span different spheres from the grassroots to the national level and have different relationships with policymakers will bring different definitions and conceptions, as well as alternative solutions, to the issue of adolescent pregnancy in the policy arena. This plurality of influence and knowledge provides a better understanding of the structure and dynamics of the public and policymakers, which is necessary for defining adolescent pregnancy in a manner that leads to its prioritization.

Second, it is critical that these communities of operatives (stakeholders) are familiar with the selection principles of public arenas and deliberately adapt their social problem claims to fit their target audience. Stakeholders should employ novel symbols to frame the importance of prioritizing adolescent pregnancy. In the 2015 visit to Kenya, President Barack Obama equated a disinvestment in adolescent girls and young women to a football team playing with only half their players [55]. Given that football is well loved in Africa, it was a relatable imagery that helped start a discussion on inclusivity of adolescents and young women.

Third, operatives can leverage already existing dominant and concurrent social problems. For example, there has been a recent focus (both technical and funding related) on adolescents who are at risk of or have HIV [56]. Given that HIV and pregnancy are both acquired through sexual intercourse, it is possible that adolescent pregnancy may gain traction by combining forces with the issue of HIV among adolescents and presenting a comprehensive construct of adolescent SRH [57]. Operatives can also anchor adolescent SRH into national and regional commitments such as increasing access to contraceptives and thus frame the issue as a foundational element not just for reproductive health, but also for social and economic equality [9,

58, 59]. For example, adolescent pregnancy resulting from unmet contraceptive need can be presented as a deterrent to national development and achieving the demographic dividend, which is a key focus of the African Union member states [59]. Thus, universal access to contraceptives (including to adolescents) can build momentum toward a demographic transition, which in turn can accelerate economic gains that benefits the society at large [8].

In conclusion, we recommend, a multidimensional problem definition, which necessitates responses on many diverse fronts, ideally simultaneously, to leverage catalytic action. For example, increasing access to high-quality healthcare, improving educational opportunities for girls, and implementing changes in laws regarding eligibility of teenagers to receive low or no cost, confidential healthcare can be overwhelming for any government to tackle, especially those with resource constraints and given the "stigma" and controversy associated with adolescent sexuality. Finally, there is need to accelerate research and innovation addressing how to improve problem definition and political prioritization not only to improve the SRH of adolescents, but also as a priority human rights and social justice issue. Key research areas of focus could include how to strengthen in-country's mechanisms to frame adolescent SRH as a priority equity issue, allocate financial resources and incentives for SRH service provision, and strengthen inter-sectoral collaborations and linkages across stakeholders.

The study limitations deserve mention. One limitation of the study is that we used purposive sampling, which has inherent problems with generalizability. In mitigation, the members interviewed were already representing their organizations and line ministries at the national technical working group for adolescent SRH and as such their views likely reflect a larger population. Secondly, we did not interview a representative from the ministry of education, which is an important arena and stakeholder of adolescents. Although we interviewed a tertiary level education sector representative, we acknowledge that a large proportion of the adolescents will be in primary or high school and that this important opinion is missed in this paper. Thirdly, interviews were conducted exclusively with national level stakeholders; therefore, sub-national variations in political prioritization in the devolved counties as well as the important voices of adolescent girls themselves are not adequately represented. Lastly the Public Arenas Model is limited in that it does not provide insight into whether a well designed adolescent SRH policy will be well implemented enough to halt and reverse the current adverse trends in adolescent pregnancy in SSA. Nevertheless, we believe that the Public Arenas Model approach provides a systematic and integrated way for different adolescent's stakeholders to think through and develop shared understandings of the problem. This systematic and shared understanding can help initiate, organize, potentially redefine and sustain adolescent pregnancy as a problem that requires political priority and resource allocation.

## Supporting information

**S1 File. Consolidated criteria for reporting qualitative research (COREQ) checklist.**
(DOC)

**S2 File. Semi-structured interview guide.**
(DOC)

## Acknowledgments

We thank the respondents who participated in this study, Professor Ruth Malone (UCSF School of Nursing) for her instruction in framing this paper, and the Director General of KEMRI for his administrative facilitation of this work.

## Author Contributions

**Conceptualization:** Maricianah A. Onono, Eric Goosby, Claire D. Brindis.

**Data curation:** Maricianah A. Onono, Claire D. Brindis.

**Formal analysis:** Maricianah A. Onono, Elizabeth A. Bukusi, Justin S. White, Claire D. Brindis.

**Investigation:** Maricianah A. Onono, George W. Rutherford, Claire D. Brindis.

**Methodology:** Maricianah A. Onono, George W. Rutherford, Justin S. White, Eric Goosby, Claire D. Brindis.

**Project administration:** Maricianah A. Onono.

**Resources:** Maricianah A. Onono.

**Software:** Maricianah A. Onono.

**Supervision:** George W. Rutherford, Elizabeth A. Bukusi, Justin S. White, Eric Goosby, Claire D. Brindis.

**Writing – original draft:** Maricianah A. Onono.

**Writing – review & editing:** Maricianah A. Onono, George W. Rutherford, Elizabeth A. Bukusi, Justin S. White, Eric Goosby, Claire D. Brindis.

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
