## [Decision Letter · Decision Letter 0]

23 Jul 2020

PONE-D-20-19160

Political prioritization and the competing definitions of adolescent pregnancy in Kenya:  An application of the Public Arenas Model

PLOS ONE

Dear Dr. Onono,

Thank you for submitting your manuscript to PLOS ONE. After careful consideration, we feel that it has merit but does not fully meet PLOS ONE’s publication criteria as it currently stands. Therefore, we invite you to submit a revised version of the manuscript that addresses the points raised during the review process.

We look forward to receiving your revised manuscript.

Kind regards,

Joshua Amo-Adjei, Ph.D

Academic Editor

PLOS ONE

Journal Requirements:

Reviewers' comments:

Reviewer's Responses to Questions

**Comments to the Author**

1. Is the manuscript technically sound, and do the data support the conclusions?

Reviewer #1: Yes

Reviewer #2: Partly

2. Has the statistical analysis been performed appropriately and rigorously? 

Reviewer #1: Yes

Reviewer #2: N/A

3. Have the authors made all data underlying the findings in their manuscript fully available?

Reviewer #1: Yes

Reviewer #2: Yes

4. Is the manuscript presented in an intelligible fashion and written in standard English?

Reviewer #1: Yes

Reviewer #2: Yes

5. Review Comments to the Author

Reviewer #1: This paper is of potential interest but needs some revision. I would like to see the authors considering the following:

1. The methods need to be explained in more detail Sufficient detail needs to be given so that another researcher could replicate the research. Details required are quality assurance of the translation of the arena model (in form of flow charts etc ) and the data set, ethical approval, protocol for participants and obtaining of informed consent, training of data collectors, and data handling and management including the prevention of data linking.

2. Analysis and integration of the themes from different sectors need to be justified – was a software necessary for the analysis?

3. Details of the piloting of the piloting need to be given.

Reviewer #2: This is a qualitative study, and does not require statistical analysis. The manuscript is presented in intelligible fashion save for structure- I am concerned about the position of theoretical underpinning in material and methods section. Standard English language has been used.

6. PLOS authors have the option to publish the peer review history of their article (what does this mean?). If published, this will include your full peer review and any attached files.

Reviewer #1: **Yes: **Gabriel O Dida

Reviewer #2: **Yes: **Kageha Emmy

---

## [Author Response · Author response to Decision Letter 0]

1 Aug 2020

31 Jul. 2020

Thank you for your time and careful review of our manuscript “Political prioritization and the competing definitions of adolescent pregnancy in Kenya: An application of the Public Arenas Model.” We deeply appreciate the feedback from the reviewers.

We have addressed your comments in the revised manuscript and have detailed our responses to each point below (in blue font). 

Thank you very much for your time and consideration of this revised manuscript for publication in PLOS ONE. If there is any additional information or details about the study that we can provide, please do not hesitate to contact me. We look forward to your response.

Sincerely,

Authors

Introduction section: 

What is the state of teenage pregnancy in Kenya? Or literature on teenage pregnancy?

Line 70-74: We have included the following two sentences: In countries such as Kenya, approximately 18% of girls between 15-19 years have begun childbearing or already have a child, and 13,000 teenage girls drop out of school every year due to pregnancy(7). If all adolescent girls completed secondary school and if adolescent mothers were employed, instead of becoming pregnant, the cumulative effect could add an estimated 3.4 billion U.S. dollars to Kenya’s gross income every year [8]. 

Clarification is needed on what "definition" precisely is in this case?

We have clarified problem definition to read as follows: 

Line 83-84: “Problem definition is the process of how networks of individuals or organizations generate consensus on what the problem is and how it should be addressed”

Page 2: Line 58-59: I wonder if it is deliberate to position population divided and leave out other outcomes, especially equality. 

We have rephrased this sentence to include gender equity—it now reads as follows 

Line 67: When given the right policies and investments, this ever-expanding youth bulge represents an opportunity to reap a demographic dividend, promote gender equity, accelerate economic growth and reduce poverty.

Methods and materials

Page: 4: Line 90-92. Did you men interpretive analysis approach or interpretive focused ethnographic approach. Interpretive ethnographic approach if performed on ethnography enquiry- grounded on (participant) observation and inquiry. 

Line 148: We have clarified this to read:“ We employed a focused ethnographic methodology to collect data for this study.”

Focused ethnography focuses on specific phenomena or shared experiences, and typically involves shorter time in the field and a smaller, often very specific, geographic area. The focused approach is made possible because we as the researchers have some background knowledge in the study area and now seeks specific information. 

Page 4: Line 103: on Theoretical underpinning. Why in methods and material section? Is this the structure of the journal? 

Line 107: We have moved this up above the methods and material section

Page 6: line 117. You mention purposive sampling: Provide clear sampling and recruitment of study participants. 

We have fleshed out this segment and it now reads as follows

Line 161-167: We used purposive and snowball sampling to identify and recruit participants. Purposive sampling was used to consciously identify potential participants. A list of potential participants was developed and prioritized according to the following criteria: job position that was previously or currently held, expected expertise and knowledge that they possess regarding SRH. Subsequently, we used snowball sampling by asking interviewees to suggest others who have contributed or influenced the processes and conduct the interviews with them. Lastly, we used theoretical sampling to select further types of interviews based on what would advance insight into the issue of problem definition.

Line 146-7: Who read through transcripts? 

Line 203-205: We have clarified this to read, “The interview transcripts were read and reread by the lead investigator and a social scientist trained in qualitative methods to identify emerging codes and categories

Page 8-9: Line 169: You have listed subjects organizations, did you get consent to include their organizations, and how does this play out in their confidentiality? A key stakeholder I expected to see here is the Ministry of Education, what is the reason for their exclusion? It would be interesting to find out how adolescent pregnancy is defined in education sector? Also excluded are donor organizations, any reason for this? 

The consent form noted that the participant names would not be included but their organisations names will be mentioned. We also mentioned that risk of confidentiality was possible but we would take extra measure. To add caution, we have removed the gender of the persons as well as the sub-departments and only mention the larger organization’s name in table 1

Ministry of education is indeed a key stakeholder but were not represented at the time. As quoted in the text, there exist deep divisions between the ministry of health and ministry of education regarding adolescent SRH. The closest representative was an educator from the Kenya Medical Training College, Nairobi and the ministry of Youth affairs. 

Findings section

Page 9-10: line 183-190: the excerpts provide different ways of delaying pregnancies: try to analyse and interpret the direct voice before moving to another section. Generally the excerpts are rich but under analysed. 

Thank you for the compliment. In this section, the quote provided is supporting the preceding discussion regarding definitions by the health arena and how these then determine the choice of interventions. The section reads as follows:

Line 229-239 “As shown in the quote below, within this arena, adolescent pregnancy is defined as being undesirable, unplanned or unwanted and is associated with major social problems, including persistent poverty, school failure, child abuse and neglect, health issues and mental health issues(28, 29) including higher risk of maternal mortality(30). This definition, therefore, emboldens the health arena’s claim for programmes and research to prevent and manage adolescent pregnancy. Such programmes focus on increasing an individual’s unfettered access to contraceptive information and expanded mix of contraceptive choices(31-39).

Page 11: 225: Culture and religious: Unless you mean traditional religions? If not, culture and religion cannot define teenage pregnancy in the same way; those are two different views. This is correct. We have corrected this to just read “cultural arena”

Old sentence: The religious and cultural arena defines adolescent pregnancy as an antisocial tendency or a problem of weak morals or agency. 

New sentence: Line 273 The cultural arena defines adolescent pregnancy as an antisocial tendency or a problem of weak morals or agency. 

In the list of subjects I have seen inter-religious representative does this subject represent the views of the cultural?

No, this representative does not represent the views of the cultural. However, he does represent both traditional and western religion 

Page 12: line 231: I suggest you include, if any, some quote from the NGOs and the CSOs to qualify your etic interpretation. 

We have removed this section, as it was an interpretation of what had been generally portrayed by the NGO and CSO representatives

Pages 18-21: from line 360: Discussion section: I was looking forward to this section, especially the intersections of arenas, and implications in furthering of understanding the dilemmas of teenage pregnancies in these arenas. In its current version, discussion is mainly a short summary and a number of propositions. You need to help readers see the broader significance of the work and how it relates to what is already known (or what we still need to understand) in this area. I suggest you make a strong argument for the policy or recommend further studies. 

In this paper, we discussed the results as we presented them in the results section. The discussion section is therefore more of a conclusion and on opportunity to chart the way forward. We however agree with you that this an opportunity to make a strong argument for the policy and recommend further studies. We have now included the following paragraph

Line 488-493: There is need to accelerate research and innovation addressing how to improve problem definition and political prioritization not only to improve the SRH of adolescents, but also as a priority human rights and social justice issue. Key research areas of focus could include how to strengthen in-country’s mechanisms to frame adolescent SRH as a priority equity issue, allocate financial resources and incentives for SRH service provision, and strengthen inter-sectoral collaborations and linkages across stakeholders. 

Reviewer #1: This paper is of potential interest but needs some revision. I would like to see the authors considering the following:

1. The methods need to be explained in more detail Sufficient detail needs to be given so that another researcher could replicate the research. Details required are quality assurance of the translation of the arena model (in form of flow charts etc.) and the data set, ethical approval, protocol for participants and obtaining of informed consent, training of data collectors, and data handling and management including the prevention of data linking.

Under data collection we state the following 

Line 192-194: We collected primary qualitative data using semi-structured in-depth interviews (IDIs). Interviews were conducted in English, lasted approximately 90 minutes, and were digitally recorded and transcribed verbatim by a masters level qualitative researcher. 

Data set is with all the anonymized transcripts is available and deposited in a public repository

All interviews were conducted in a private location at the participant’s discretion by a trained and experienced qualitative researcher (more details in consolidated guidelines (COREQ) for reporting qualitative research (S1 File).

Line 221-227 The research was reviewed and approved by the Scientific and Ethics Review Unit (SERU Study 3738) at the Kenya Medical Research Institute (KEMRI) and the Committee for Human Research of the University of California, San Francisco (UCSF). All participants (table 2) provided written informed consent prior to the interview being conducted. The digital audio recording of the in-depth interviews were not initiated until after the informed consent process was complete, the participant had agreed to the recording, and any initial introductions that might include identifying information had been completed. Participants were not reimbursed for participating in the study.

2. Analysis and integration of the themes from different sectors need to be justified – was a software necessary for the analysis?

Line 196-197: The data were analyzed using a theory-informed thematic analytical approach (25) using Dedoose qualitative software. 

3. Details of the piloting of the piloting need to be given.

No piloting was done. We used a questionnaire that had been previously used in this setting 

Reviewer #2: This is a qualitative study, and does not require statistical analysis. The manuscript is presented in intelligible fashion save for structure- I am concerned about the position of theoretical underpinning in material and methods section. Standard English language has been used.

We have moved this section to be above the materials and methods section

---

## [Editor Report · Decision Letter 1]

6 Aug 2020

PONE-D-20-19160R1

Political prioritization and the competing definitions of adolescent pregnancy in Kenya:  An application of the Public Arenas Model

PLOS ONE

Dear Dr. Onono,

Thank you for submitting your manuscript to PLOS ONE. After careful consideration, we feel that it has merit but does not fully meet PLOS ONE’s publication criteria as it currently stands. Therefore, we invite you to submit a revised version of the manuscript that addresses the points raised during the review process.

We look forward to receiving your revised manuscript.

Kind regards,

Joshua Amo-Adjei, Ph.D

Academic Editor

PLOS ONE

Additional Editor Comments (if provided):

That no participants were drawn from the Ministry of Education is a key limitation to your study and this should be pointed out as such. 

---

## [Author Response · Author response to Decision Letter 1]

6 Aug 2020

6 August. 2020

Thank you for your time and careful review of our manuscript “Political prioritization and the competing definitions of adolescent pregnancy in Kenya: An application of the Public Arenas Model.” We appreciate the feedback.

We have addressed your comment in the revised manuscript and have detailed our response to the query below (in blue font). 

Thank you very much for your time and consideration of this revised manuscript for publication in PLOS ONE. If there is any additional information or details about the study that we can provide, please do not hesitate to contact me. We look forward to your response.

Sincerely,

Authors

Additional Editor Comments (if provided):

That no participants were drawn from the Ministry of Education is a key limitation to your study and this should be pointed out as such. 

We have now included this in Line 396-400: Secondly, we did not interview a representative from the ministry of education, which is an important arena and stakeholder of adolescents. Although we interviewed a tertiary level education sector representative, we acknowledge that a large proportion of the adolescents will be in primary or high school and that this important opinion is missed in this paper.

---

## [Editor Report · Decision Letter 2]

11 Aug 2020

Political prioritization and the competing definitions of adolescent pregnancy in Kenya:  An application of the Public Arenas Model

PONE-D-20-19160R2

Dear Dr. Onono,

We’re pleased to inform you that your manuscript has been judged scientifically suitable for publication and will be formally accepted for publication once it meets all outstanding technical requirements.

Kind regards,

Joshua Amo-Adjei, Ph.D

Academic Editor

PLOS ONE
---

## [Editor Report · Acceptance letter]

3 Sep 2020

PONE-D-20-19160R2 

Political prioritization and the competing definitions of adolescent pregnancy in Kenya: An application of the Public Arenas Model 

Dear Dr. Onono:

I'm pleased to inform you that your manuscript has been deemed suitable for publication in PLOS ONE. Congratulations! Your manuscript is now with our production department. 

Kind regards, 

on behalf of

Dr. Joshua Amo-Adjei 

Academic Editor

PLOS ONE